# Isolation and Characterization of a Native *Metarhizium rileyi* Strain Mrpgbm2408 from *Paralipsa gularis* in Maize: First Data on Efficacy and Enzymatic Host Response Dynamics

**DOI:** 10.3390/insects16090872

**Published:** 2025-08-22

**Authors:** Yunhao Yao, Kaiyu Fu, Xiaoyu Wang, Guangzu Du, Yuejin Peng, Guy Smagghe, Wenqian Wang, Bin Chen

**Affiliations:** 1College of Plant Protection, Yunnan Agricultural University, Kunming 650201, China; yaoyunh@163.com (Y.Y.); echotangerine@163.com (K.F.); 18087845021@163.com (X.W.); 15288457381@163.com (G.D.); 2021053@ynau.edu.cn (Y.P.); 2Institute of Entomology, Guizhou University, Guiyang 550025, China; guysma9@gmail.com; 3Department of Biology, Vrije Universiteit Brussel (VUB), 1050 Brussels, Belgium

**Keywords:** *Paralipsa gularis*, *Metarhizium rileyi*, entomopathogenic fungus, biological characteristics, stress resistance

## Abstract

In this study, we report, for the first time, the agricultural pest *Paralipsa gularis* naturally infected with the fungus *Metarhizium* sp. Molecular analysis using ITS and EF-1α sequences identified the strain as *Metarhizium rileyi*, which we named *M. rileyi* Mrpgbm2408. Laboratory tests showed that the strain grew best on SMAY medium under continuous light at 25 °C. It also displayed strong virulence against *P. gularis*. We found that protective and detoxifying enzymes in the pest played a role in defending against infection by *M. rileyi* Mrpgbm2408. These findings highlight the potential of this fungal strain as a biological control agent, offering promising implications for reducing crop losses, lowering economic damage, and supporting environmental sustainability.

## 1. Introduction

*Paralipsa gularis* (Zeller) (Lepidoptera: Pyralidae), a moth species native to Southeast Asia, has emerged as a significant pest in both agricultural and storage environments. With the rapid expansion of international food trade, this species has dispersed across Asia, Northern Europe, and North America, and is now widely distributed throughout many provinces in China [1,2,3]. In recent years, shifts in cropping systems and climate patterns have expanded its ecological niche from grain storage facilities into field ecosystems. Notably, *P. gularis* has become a serious threat to maize production, causing tassel rot and contributing to substantial yield losses [4,5]. Given its growing agricultural impact, effective and sustainable control of *P. gularis* is vital for ensuring food security.

Conventional pest control strategies rely on chemical, physical, and biological approaches [6], with chemical pesticides remaining the predominant choice due to their rapid efficacy. However, intensive and prolonged pesticide use has accelerated the development of insecticide resistance through genetic adaptation in pest populations [7,8,9]. Moreover, chemical inputs can negatively impact non-target organisms, including pollinators and natural enemies, while contributing to environmental contamination and biodiversity loss [10,11]. In this context, biological control using entomopathogenic fungi offers a promising and environmentally sustainable alternative.

Among entomopathogenic fungi, species of the genus *Metarhizium* are notable for their natural pathogenicity to a wide range of insect hosts. Several *Metarhizium* strains have already been formulated into commercial biopesticides and successfully integrated into pest management programs [12]. Their effectiveness is attributed to traits such as host specificity, environmental adaptability, and persistence in agroecosystems. However, their practical use can be constrained by abiotic stresses, such as UV exposure, temperature extremes, and nutrient limitations, which may reduce fungal survival, infectivity, and virulence [13]. A deeper understanding of these environmental influences is essential for improving both the field performance and formulation stability of fungal biopesticides.

Insects defend against microbial invasion by activating innate immune responses, including the regulation of protective and detoxification enzyme systems. Pathogens, in turn, employ strategies to suppress or evade these defenses to establish infection [14,15,16,17]. Antioxidant enzymes such as superoxide dismutase (SOD), peroxidase (POD), and catalase (CAT) mitigate oxidative damage, while detoxification enzymes, including cytochrome P450 monooxygenases (P450), carboxylesterases (CarE), acetylcholinesterases (AChE), and glutathione *S*-transferases (GSTs), process and neutralize toxic compounds through hydrolysis, redox, and conjugation reactions [18,19,20,21,22]. The modulation of these enzyme systems is central to insect defense and resistance, yet the physiological responses of *P. gularis* to entomopathogenic fungal infection remain poorly understood.

In this study, we isolated and characterized a novel *Metarhizium* sp. strain from naturally infected *P. gularis* larvae collected in Longling County, Baoshan City, Yunnan Province, China. Using morphological examination and ITS-rDNA sequencing, we identified the fungal strain and evaluated its biological characteristics, virulence, and infection mechanisms. Our findings provide new insights into host–pathogen interactions and highlight the potential of *Metarhizium* sp. as an innovative biological control agent (BCA), contributing to the development of sustainable and ecologically sound strategies for managing *P. gularis* populations in agricultural ecosystems.

## 2. Materials and Methods

### 2.1. Test Larvae and Strains

Infected and healthy *P. gularis* larvae were collected from a cornfield in Bengmiao Village, Longling County, Baoshan City, Yunnan Province, China (98°78′91″ E, 24°53′50″ N, H ≈ 1864). Healthy larvae were reared using corn cobs in the Insect Pathogen Laboratory of the College of Plant Protection at Yunnan Agricultural University.

### 2.2. Strain Isolation and Purification

Infected *P. gularis* larvae were placed in sterile 90 mm Petri dishes and incubated in a humid environment at 25 °C. Once fungal spores germinated, a small number were collected using an inoculation needle and transferred to Sabouraud dextrose agar plus yeast extract (SDAY) medium under sterile conditions. The cultures were incubated at 25 ± 1 °C, 75 ± 5% relative humidity (RH), with a 12:12 light–dark (L:D) photoperiod; Mycelia were transferred to fresh media for purification [23]. A single strain with consistent morphology was selected, designated Mrpgbm2408, and stored at 4 °C for future use.

### 2.3. Molecular Identification of the Strain

Genomic DNA of strain Mrpgbm2408 was extracted using the DZUP Genomic DNA Extraction Kit (Shanghai Sangong Bioengineering, Shanghai, China). The internal transcribed spacer (ITS) region was amplified using universal primers ITS1 (5′-TCCGTAGGTGAACCTGCGG-3′) and ITS4 (5′-TCCTCCGCTTATTGATATGC-3′); The EF1-alpha region was amplified using universal primers 2218R (5′-ATGACACCRACRGCRACRGTYTG-3′) and 983F (5′-GCYCCYGGHCAYCGTGAYTTYAT-3′) [24]. The PCR mixture (25 μL) contained 1 μL of each primer, 1 μL of DNA template, 12.5 μL of 2 × Taq PCR Master Mix, and 9.5 μL of ddH_2_O. PCR conditions were 98 °C for 2 min; 35 cycles of 98 °C for 15 s, 50 °C for 25 s, 72 °C for 30 s; and a final extension at 72 °C for 10 min. Products were visualized by 0.8% agarose gel electrophoresis. Samples with clear bands were submitted for sequencing.

### 2.4. Culture Characteristics of Strain Mrpgbm2408

Strain Mrpgbm2408 (1 × 10^7^ conidia/mL) was inoculated onto 90 mm plates with Potato Dextrose Agar (PDA), SDAY, and Sabouraud maltose agar plus yeast extract (SMAY) via the spot inoculation method and incubated at 25 ± 1 °C, 70 ± 5% RH, under constant light (L:D = 24:0). Colony morphology and color were recorded over 4–5 days. Mycelia and spores were collected for the microscopic observation of conidiophore morphology [25] and stored at Yunnan Agricultural University.

### 2.5. Biological and Growth Phenotype of Strain Mrpgbm2408

Following methods from Xue et al. [26] and Wang et al. [27], the effects of different media, temperatures, and photoperiods on the growth, sporulation, heat tolerance, and UV resistance of Mrpgbm2408 were assessed. Media with single carbon or nitrogen sources were used to evaluate nutritional response.

A 5 mm fungal plug from a 10-day-old culture was transferred to PDA, SDAY, and SMAY plates and incubated under the same conditions as in Section 2.4. To test temperature effects, plugs were incubated at 20 °C, 25 °C, and 30 °C. To test photoperiod effects, plugs were incubated under three light regimes: L:D = 24:0, 12:12, and 0:24. From day 3, colony diameters were measured daily using the cross method. After 15 days, spores were harvested by punching colony centers and mixing in a 0.05% Tween-80 solution. Spore concentration was calculated using a hemocytometer. Each treatment was repeated three times.

For heat tolerance, conidia (1 × 10^7^/mL) were exposed to 30 °C, 40 °C, 50 °C, 60 °C, and 70 °C in a water bath for 2 min, then inoculated into 100 mL of SDY medium. For UV resistance, spore plates were irradiated with 25 W UV lamps (40 cm distance) for 2, 6, and 10 min. Spores were then transferred to SDY to evaluate germination.

Add 3% to each carbon source (fructose, sucrose, trehalose, lactose, and maltose) and add 0.3% to each nitrogen source (NH_4_Cl, gelatin, peptone, and NaNO_3_) were used to assess fungal growth by measuring colony diameters after 7 days at 25 °C. Czapek-Dox agar (CZA) served as the defined control medium.

### 2.6. Virulence of the Strain

Virulence of Mrpgbm2408 against 3rd-instar *P. gularis* larvae was evaluated using the immersion method. Larvae were dipped for 10 s in suspensions of five conidial concentrations of Mrpgbm2408: 5 × 10^8^, 5 × 10^7^, 5 × 10^6^, 5 × 10^5^, 5 × 10^4^ conidia/mL. Control larvae were treated with 0.05% Tween-80. Treated larvae were placed in 12-well plates (20 larvae per treatment, 3 replicates) and incubated at 25 ± 1 °C, 75 ± 5% RH, with a L:D = 16:8 photoperiod. Mortality was recorded daily for 10 days. Larvae were considered dead if unresponsive to gentle brush stimulation. Mortality, corrected mortality, and cumulative mortality were calculated.
$$Mortality\;rate\;(\%)=\frac{Number\;of\;dead\;insects}{Number\;of\;test\;insects}\times 100$$

$$Corrected\;mortality\;rate\;(\%)=\frac{Treatment\;group-control\;group}{1-control\;group}\times 100$$

$$Cumulative\;mortality\;(\%)=\frac{Number\;of\;dead\;insects}{Total\;number\;of\;insects\;tested}\times 100$$


### 2.7. Protective and Detoxification Enzymes in P. gularis

Third-instar larvae were treated with a lethal concentration of Mrpgbm2408 conidia and sampled at 12, 24, 36, 48, and 60 h post-treatment. Controls were treated with 0.05% Tween-80. Ten larvae per time point were used, with three replicates per treatment. Larvae were flash-frozen in liquid nitrogen and stored at −80 °C for enzyme analysis.

Larvae were rinsed with saline, dried, and homogenized in pre-cooled mortars. Homogenates were centrifuged at 8000× *g* for 10 min at 4 °C. Supernatants were collected for enzyme assays. Protective enzymes (SOD, CAT, and POD) and detoxification enzymes (CarE, AChE, and GST) were measured using Boxbio kits; cytochrome P450 was measured using kits from Jiangsu Jingmei Biological Technology (Yancheng, China).

### 2.8. Data Analysis

All experimental data, including larval mortality, enzyme activities, and fungal growth parameters, were analyzed using SPSS 25.0 software for one-way or two-way analysis of variance (ANOVA), and multiple comparisons of means were performed using Tukey’s HSD test. NJ-method phylogenetic analysis was conducted using MEGA7. Figures were created using GraphPad Prism 9.5.0 and Adobe Illustrator 2023.

## 3. Results

### 3.1. Identification of the Isolate as Metarhizium rileyi

Infected *P. gularis* larvae were initially covered in white mycelium, later producing dark green spores (Figure 1A). Growth on PDA, SDAY, and SMAY media displayed distinct colony morphologies. Notably, SMAY supported the fastest sporulation (starting at day 8) and robust colony development, while PDA and SDAY showed delayed and less uniform growth (Figure 1(Ba–Bf)). Microscopic examination revealed characteristic conidiophores and ovoid-to-pointed conidia (Figure 1(Bg–Bi)). ITS sequence analysis confirmed close homology to *M. rileyi* (MH893365.1) (Figure 1C) and EF-1α sequence analysis confirmed close homology to *M. rileyi* (MH986285.1) (Figure 1D), validating the morphological identification of strain Mrpgbm2408.

### 3.2. Culture Medium Influences Growth, Sporulation, and Stress Resistance

Over 15 days, SMAY medium supported the highest growth rate (1.52 ± 0.26 mm/d) and spore density (3.77 × 10^7^ spores/mm^2^), outperforming both SDAY and PDA (*p* < 0.01; Figure 2A). UV exposure (25 W) further highlighted SMAY’s advantage, maintaining higher germination after 2 min and significantly outperforming others after 6 min (Figure 2B). Under thermal stress (30 °C and 60 °C), spores from PDA exhibited significantly reduced viability, namely 42% germination at 30 °C and near-zero at 60 °C, compared with SDAY and SMAY (Figure 2C).

### 3.3. Temperature Influences Growth, Sporulation, and Stress Resistance

Cultures grown at 25 °C displayed the most robust development, with a growth rate of 1.51 ± 0.27 mm/d and spore production of 3.35 × 10^7^ spores/mm^2^, significantly higher than at 20 °C or 30 °C (Figure 3A). Under UV stress, 20 °C cultured spores were most sensitive, with significant decline even after 2 min irradiation (*p* < 0.01; Figure 3B). While 30 °C water baths did not impair viability across treatments (77–78% germination), 60 °C treatments yielded extreme inhibition (≤7.4%; Figure 3C).

### 3.4. Photoperiod Influences Growth, Sporulation, and Stress Resistance

Continuous light (L:D = 24:0) resulted in the fastest growth (1.47 ± 0.39 mm/d) and maximum sporulation (3.82 × 10^7^ spores/mm^2^), compared with L:D = 12:12 or full darkness (Figure 4A). Light-exposed spores exhibited superior resistance to heat and UV; for example, 24L:0D spores maintained 64.9% germination after 2 min UV, significantly higher than dark-grown spores (Figure 4B,C).

### 3.5. Nutrient Source Modulates Fungal Growth

Colonial morphology varied with carbon and nitrogen sources (Figure 5). Notably, maltose enhanced growth compared with sucrose, lactose, fructose, and trehalose, while peptone was the most effective nitrogen source, significantly increasing colony size relative to NaNO_3_, gelatin, or NH_4_Cl.

### 3.6. Pathogenicity Against P. gularis Larvae

Mrpgbm2408 exhibited strong pathogenicity in insect bioassays. Cumulative larval mortality reached 82% by day 10 at 5 × 10^8^ conidia/mL (Figure 6). The calculated LC_50_ value was 2.28 × 10^6^ conidia/mL at day 10. The LT_50_ was 5.3 days with 5 × 10^8^ conidia/mL, and 7.1 days with 5 × 10^7^ conidia/mL (Table 1). These insect toxicity data confirm a strong pathogenicity with low LC_50_ and LT_50_ values against *P. gularis* larvae.

### 3.7. Host Enzyme Responses to Infection

Infection by Mrpgbm2408 significantly altered host enzyme activities. Specifically, SOD declined until 24 h post-infection, followed by recovery by 60 h; it was lowest at 12 h (1.61 U/g; *p* < 0.05) (Figure 7A). POD peaked at 24 h (5470.7 U/g; 5.7× control; *p* < 0.05), staying elevated throughout (Figure 7B). Figure 7C shows that CAT reached its highest activity at 48 h (2744.6 U/g; 6.7× control; *p* < 0.05).

On the detoxification enzymes, GST increased steadily until 60 h (0.88 U/g; 6.8× control; *p* < 0.05) (Figure 8A), AChE peaked at 36 h (3675.6 U/g; 4.4× control; *p* < 0.05) (Figure 8B), and CarE rose gradually to 60 h (35.6 U/g; 2× control); a significant difference was observed at 24 h (Figure 8C), and CYP450 showed a transient increase, peaking at 24 h (6.66 pmol/L; 1.6× control; *p* < 0.05), and then declining by 48 h (Figure 8D).

Taken together, the observed enzyme dynamics underscore a complex host response, with an initial suppression of antioxidative defense followed by compensatory activation of detoxification mechanisms. Notably, these host biochemical responses reveal potential enzymatic biomarkers of infection and resistance pathways.

## 4. Discussion

This study provides the first report of *M. rileyi* strain Mrpgbm2408 isolated from naturally infected *Paralipsa gularis* in the field, thereby expanding our understanding of the host range and ecological adaptability of this entomopathogenic fungus. Morphological characteristics, supported by ITS and EF-1α rDNA sequencing, confirmed the identity of the isolate as *M. rileyi*, a species recognized for its high virulence and environmental resilience in pest control applications [23,26].

Our findings show that culture medium composition significantly influences the physiological performance of *M. rileyi*. The optimal conditions for Mrpgbm2408 growth were SMAY medium, a temperature of 25 °C, and continuous light. Among PDA, SDAY, and SMAY, the latter supported the highest growth rate and conidial yield, likely due to its enriched nutritional profile, particularly its higher peptone content. This observation aligns with previous findings [27,28] and underscores the importance of medium formulation in optimizing fungal production for commercial bioinsecticides. Moreover, spores produced on SMAY exhibited superior resistance to UV and heat stress, a trait valuable for large-scale formulations where persistence under field conditions is critical. Optimizing formulations, such as incorporating UV protectants or oil-based carriers, may further enhance practical application.

Temperature and photoperiod were also key determinants of fungal development. Optimal growth and sporulation at 25 °C are consistent with earlier reports [29,30]. Continuous light (L:D = 24:0) significantly enhanced vegetative growth and sporulation, likely reflecting long-term adaptation to the high-sunlight conditions of Baoshan, Yunnan. Conidia produced under these optimal conditions showed high thermotolerance and UV resistance, traits essential for field applications where abiotic stresses often limit fungal efficacy [31,32]. Nevertheless, extensive field trials remain necessary to evaluate the persistence, virulence, and ecological impact of Mrpgbm2408 across variable agroecological settings.

Pathogenicity assays confirmed the high virulence of *M. rileyi* Mrpgbm2408, with corrected mortality exceeding 80% at 5 × 10^8^ conidia/mL and an LT_50_ of 5.3 days. These results highlight its strong potential as a BCA for *P. gularis*, a pest of growing concern in stored grain environments [5]. The effective control achieved at relatively low inoculum concentrations suggests that this strain could be an efficient component of integrated pest management (IPM) programs. Expanding host range assessments will help determine whether Mrpgbm2408 can target other economically important lepidopteran pests.

Infection with Mrpgbm2408 triggered distinct biochemical changes in *P. gularis* larvae. When infested by fungi, the oxygen homeostasis in *P. gularis* is disrupted and a large amount of reactive oxygen species accumulates in the body, which stimulates antioxidant enzyme responses [17,18]. SOD activity decreased post-infection, suggesting an overwhelmed oxidative stress response, while POD and CAT were significantly upregulated, which are likely compensatory mechanisms to detoxify hydrogen peroxide and limit oxidative damage. However, with the growth of infection time, the activity of the protective enzymes gradually returned to the early stage of infection, which also indicated that they assisted the host in surviving the stress period. These findings are consistent with earlier observations in other insect–fungus systems [33,34] and highlight the importance of oxidative stress in fungal pathogenicity.

Detoxification enzymes (CarE, AChE, GST, and CYP450) displayed a pattern of early activation followed by decline, indicating an initial host defense response that weakened as infection progressed and fungal toxins accumulated. GST activity peaked under LC_50_ exposure, consistent with previous reports of its role in early-stage antifungal defense [35]. Differences in AChE and CYP450 responses compared with other host–pathogen systems, such as *Spodoptera frugiperda* and *Aphis citricola*, may reflect species-specific metabolic and neurophysiological adaptations to fungal infection [36,37,38].

The combined traits of high virulence, environmental resilience, and adaptability to nutritional and climatic factors make *M. rileyi* Mrpgbm2408 a promising candidate for the biological control of *P. gularis*. Its robust conidia, capable of withstanding UV and heat stress, suggest good field performance, particularly in subtropical monsoon climates like Yunnan, which closely match its native habitat.

Finally, the observed enzymatic disruptions in *P. gularis* provide valuable insights into fungal pathogenicity and host susceptibility. These biochemical responses may serve as early biomarkers of infection or as targets for synergistic pest management strategies involving other microbial or botanical agents. Future comparative genomics and transcriptomics could help identify key genes associated with virulence, environmental adaptation, and host specificity. Additionally, combining Mrpgbm2408 with complementary microbial agents or botanicals could yield synergistic effects and slow resistance development.

## 5. Conclusions

This study identified and characterized a novel strain of *M. rileyi* with strong insecticidal activity against *P. gularis* and notable environmental resilience. The optimum culture conditions for indoor cultivation of this strain were SMAY medium, 25 °C, and L:D = 24:0. Protective and detoxifying enzymes in *P. gularis* play important defense roles in its resistance to *M. rileyi* Mrpgbm2408 infestation, highlighting the great potential of *M. rileyi* Mrpgbm2408 strain as a biocontrol agent. The optimization of culture conditions, together with detailed insights into host–pathogen interactions, provides a solid foundation for developing effective and sustainable fungal biopesticides. Future research should focus on formulation strategies to enhance spore stability, large-scale field evaluations under diverse agroecological conditions, and transcriptomic analyses to uncover genetic determinants of virulence and environmental adaptation, thereby maximizing the biocontrol potential of this promising strain.

## Figures and Tables

**Figure 1 insects-16-00872-f001:**
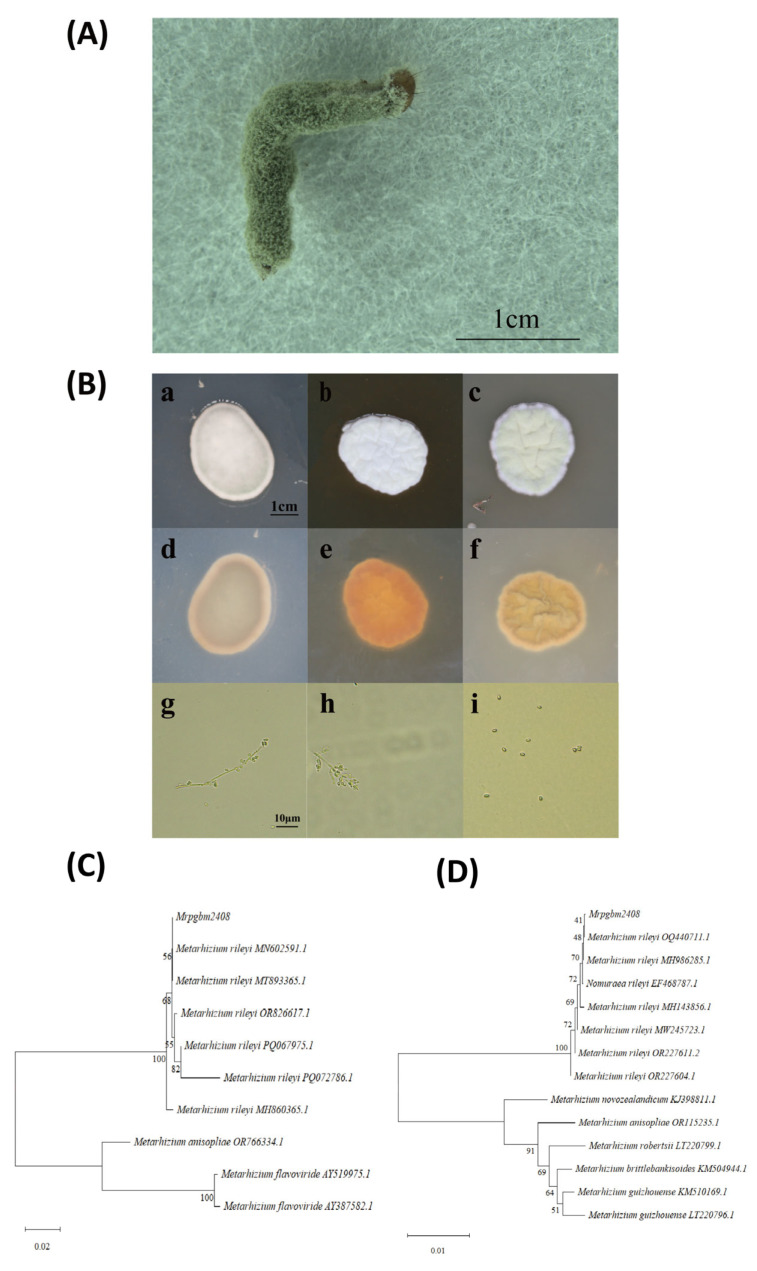
Characterization of the native *M. rileyi* strain Mrpgbm2408: (**A**) Insecticidal symptoms on 3rd-instar larvae of *P. gularis*. (**B**) Culture characteristics and spore morphology, (**a**): PDA medium front, (**b**): SDAY medium front, (**c**): SMAY medium front, (**d**): PDA medium back, (**e**): SDAY medium back, and (**f**): SMAY medium back. (**C**) Phylogenic tree of the ITS sequence of *M. rileyi* Mrpgbm2408. (**D**) Phylogenic tree of the EF1-alpha sequence of *M. rileyi* Mrpgbm2408.

**Figure 2 insects-16-00872-f002:**
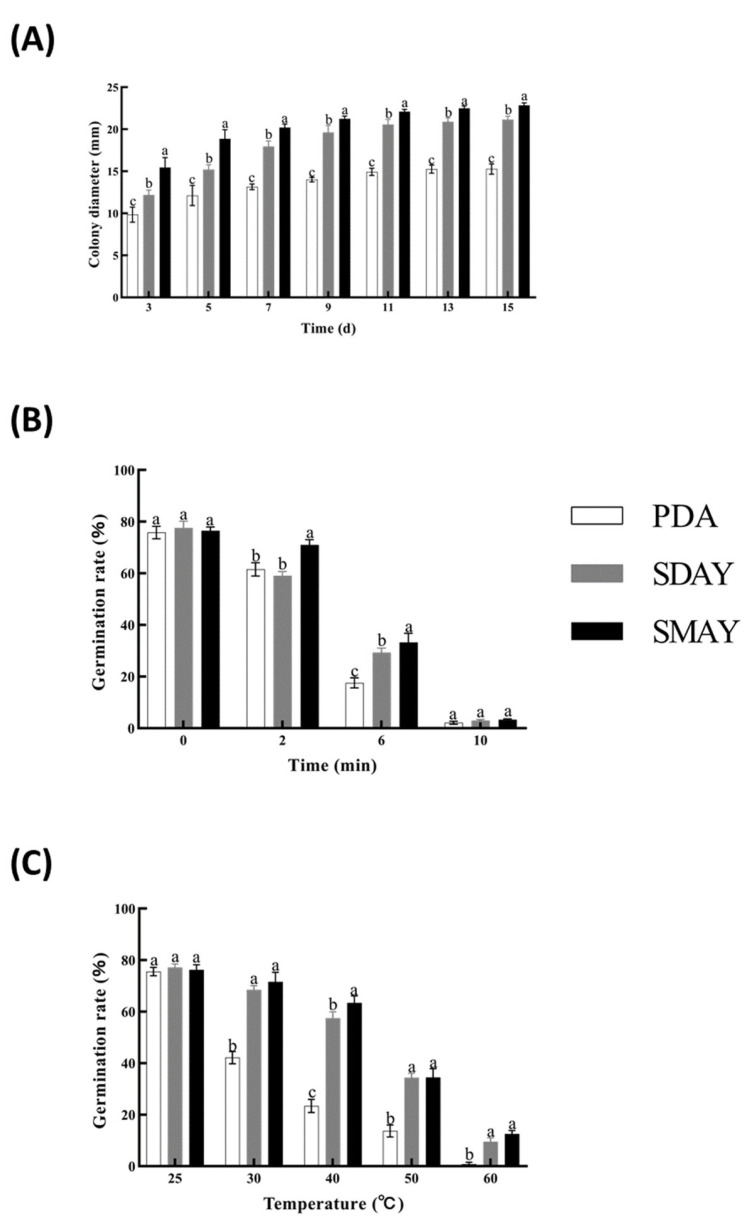
Effect of culture medium (PDA, SDAY, and SMAY) on growth and germination of the native strain of *M. rileyi*, Mrpgbm2408. Measurement of fungal growth, expressed as colony diameter, up to 15 days of culture (**A**), and effect on its resistance, expressed as percentage of germination, upon exposure to UV light for different times (0, 2, 6, and 10 min) (**B**), and exposure to different temperatures (30, 40, 50, and 60 °C, compared with the normal condition at 25 °C) (**C**). Data are expressed as mean ± SD based on three repetitions. Significant differences (*p* < 0.05) between treatment groups are indicated by different letters.

**Figure 3 insects-16-00872-f003:**
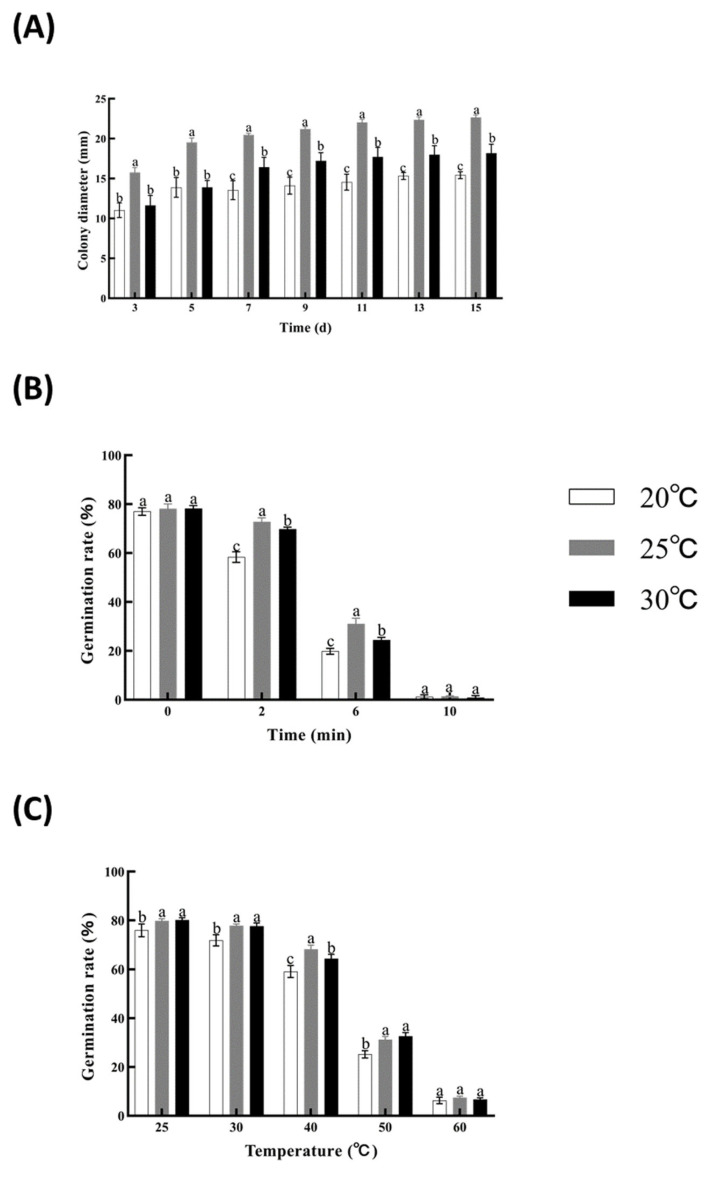
Effect of temperature (20, 25, and 30 °C) on growth and germination of the native strain of *M. rileyi*, Mrpgbm2408. Measurement of fungal growth, expressed as colony diameter, up to 15 days of culture (**A**); effect on its resistance, expressed as a percentage of germination, upon exposure to UV light for different times (0, 2, 6, and 10 min) (**B**); and exposure to different temperatures (30, 40, 50, and 60 °C, compared with the normal condition at 25 °C) (**C**). Data are expressed as mean ± SD based on three repetitions. Significant differences (*p* < 0.05) between treatment groups are indicated by different letters.

**Figure 4 insects-16-00872-f004:**
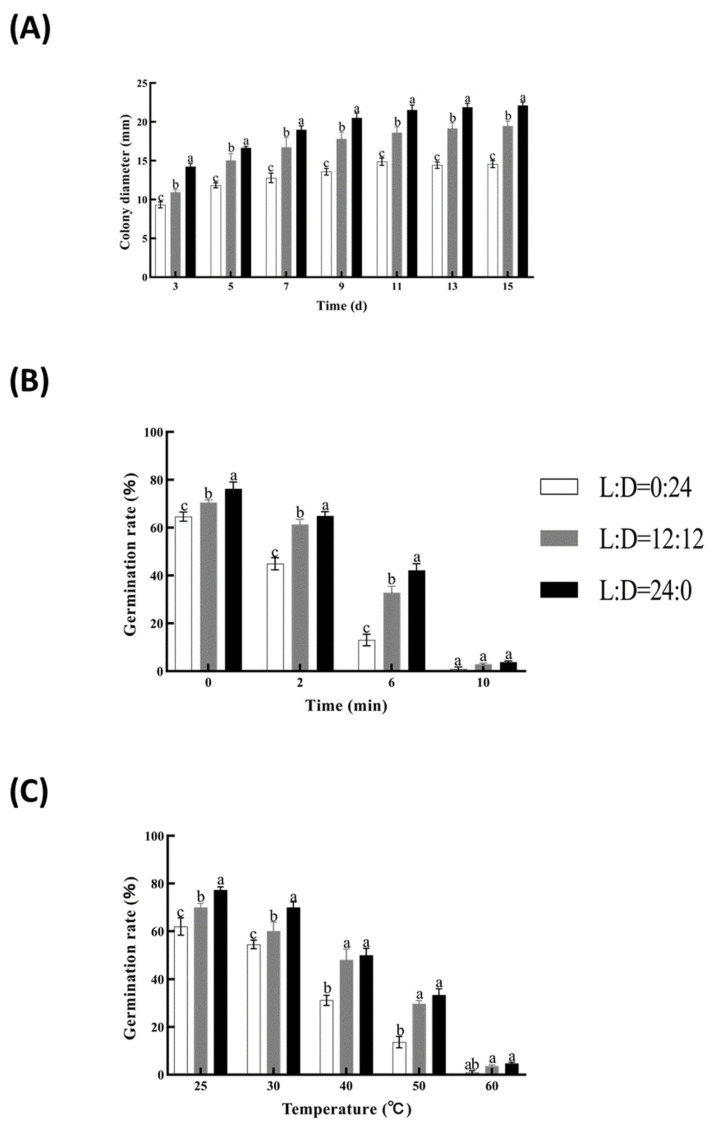
Effect of photoperiod (0:24 = L:D, 12:12 = L:D, and 24:0 = L:D) on growth and germination of the native strain of *M. rileyi*, Mrpgbm2408. Measurement of fungal growth, expressed as colony diameter, up to 15 days of culture (**A**); effect on its resistance, expressed as a percentage of germination, upon exposure to UV light for different times (0, 2, 6, and 10 min) (**B**); and exposure to different temperatures (30, 40, 50, and 60 °C, compared with the normal condition at 25 °C) (**C**). Data are expressed as mean ± SD based on three repetitions. Significant differences (*p* < 0.05) between treatment groups are indicated by different letters.

**Figure 5 insects-16-00872-f005:**
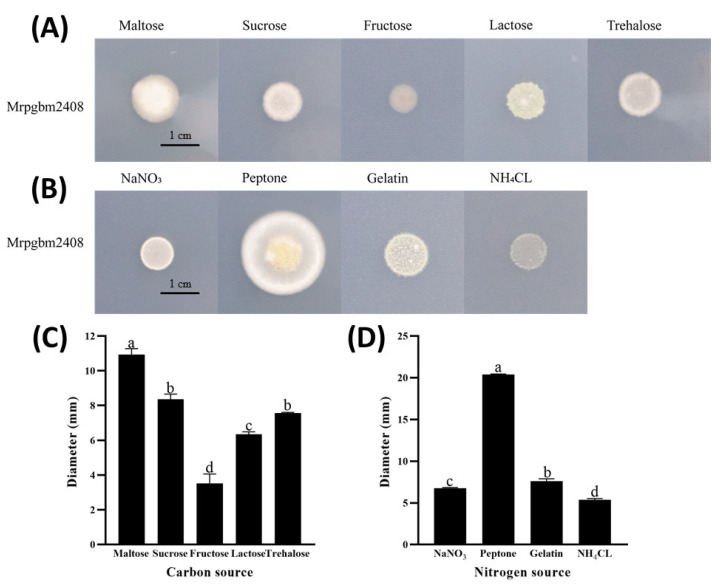
Effect of nutrient source with carbon on growth of the native strain of *M. rileyi*, Mrpgbm2408. Micrograph of fungal growth (**A**), nitrogen sources (**B**), and measurement of colony diameter (mm) when grown on different carbon (**C**) and nitrogen sources (**D**). Data are expressed as mean ± SD based on three repetitions. Significant differences (*p* < 0.05) between treatment groups are indicated by different letters.

**Figure 6 insects-16-00872-f006:**
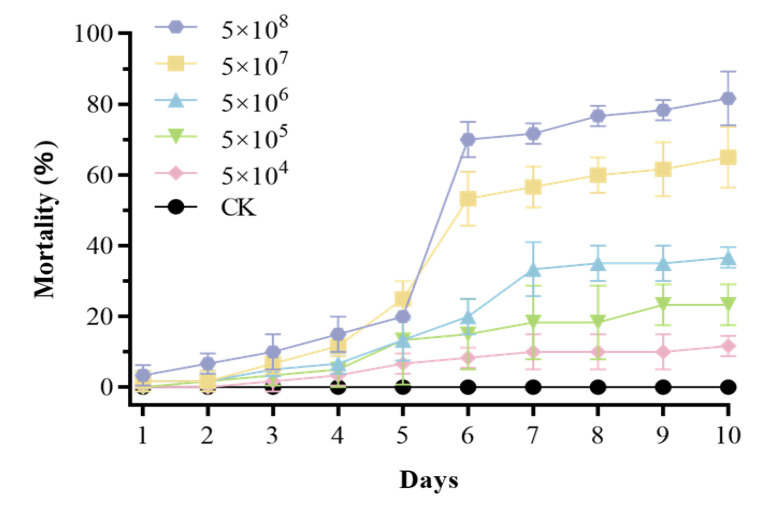
Cumulative mortality of the native strain of *M. rileyi*, Mrpgbm2408, on 3rd-instar larvae of *P. gularis* when treated at different concentrations of conidia/mL over a period of 10 days.

**Figure 7 insects-16-00872-f007:**
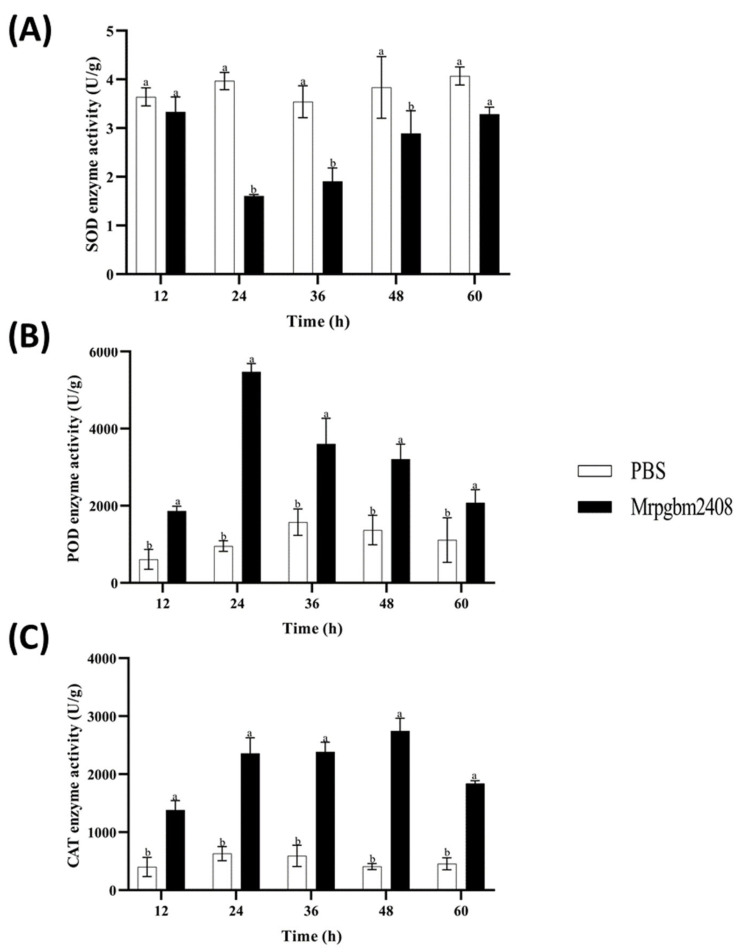
Protective enzyme activities of superoxide dismutase (SOD) (**A**), peroxidase (PER) (**B**), and catalase (CAT) (**C**) in 3rd-instar larvae of *P. gularis* upon infection with the native strain of *M. rileyi*, Mrpgbm2408, as compared with untreated controls. Data are expressed as mean ± SD based on three repetitions. Different letters between treatment and control indicate a significant difference at *p* < 0.05. Phosphate-Buffered Saline (PBS).

**Figure 8 insects-16-00872-f008:**
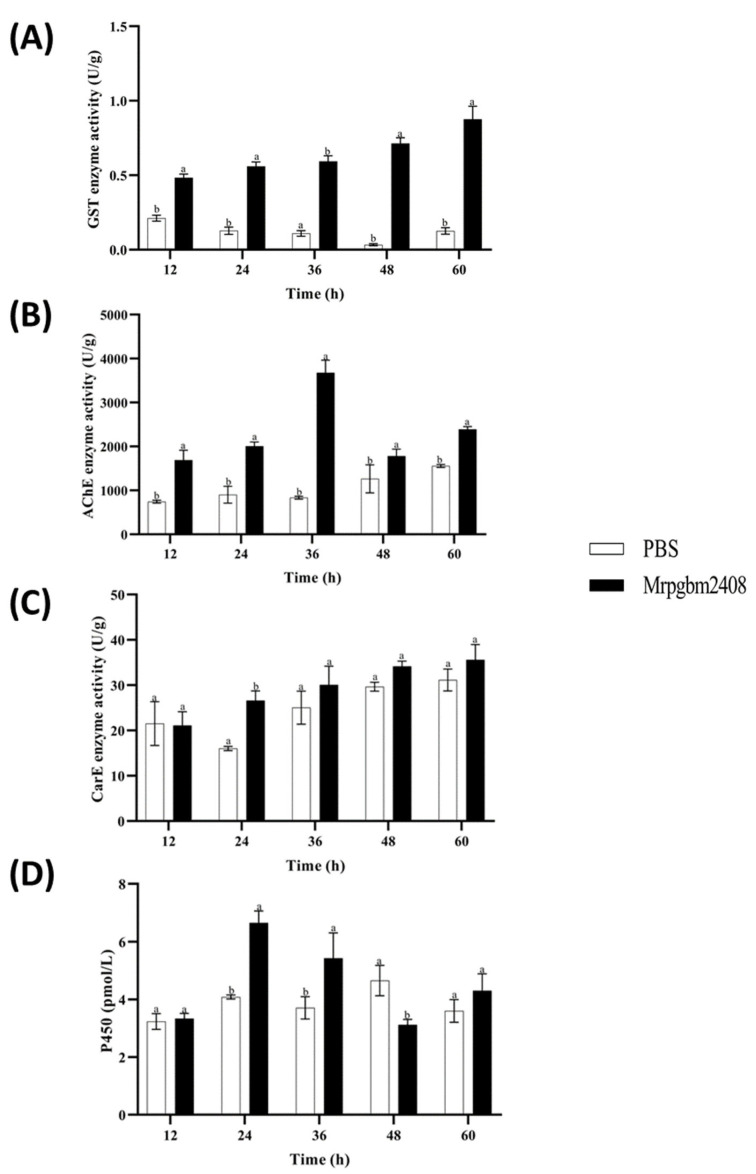
Detoxification enzyme activities of glutathione *S*-transferases (GSTs) (**A**), acetylcholinesterases (AChE) (**B**), carboxylesterases (CarE) (**C**), and cytochrome P450 monooxygenases (P450) (**D**) in 3rd-instar larvae of *P. gularis* upon infection with the native strain of *M. rileyi*, Mrpgbm2408, as compared with untreated controls. Data are expressed as mean ± SD based on three repetitions. Different letters between treatment and control indicate a significant difference at *p* < 0.05. PBS = Phosphate-Buffered Saline.

**Table 1 insects-16-00872-t001:** Toxicity of the native strain of *M. rileyi*, Mrpgbm2408, against 3rd-instar larvae of *P. gularis* with calculation of median LT and LC values for different concentrations of conidia/mL.

Conidia/mL	Regression Equation	Correlation Coefficient	LT_50_ (d)	LC_50_ (Conidia/mL)
5 × 10^8^	Y = 10.67X − 15.33	0.8660	5.3	1.36 × 10^7^
5 × 10^7^	Y = 8.606X − 13.00	0.8840	7.1	
5 × 10^6^	Y = 4.758X − 7.333	0.8707	-	
5 × 10^5^	Y = 2.899X − 3.778	0.6239	-	
5 × 10^4^	Y = 1.444X − 1.778	0.6585	-	

Note: Y in the equation is the odds value, and X is the logarithm of time. “-” indicates that the mortality rate of treated *P. gularis* 3rd-instar larvae was below 50% for 10 d, and no estimate of LT_50_ could be made.

## Data Availability

The original contributions presented in this study are included in the article/supplementary material. Further inquiries can be directed to the corresponding authors.

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
