# Peer review of "Isolation and Characterization of a Native Metarhizium rileyi Strain Mrpgbm2408 from Paralipsa gularis in Maize: First Data on Efficacy and Enzymatic Host Response Dynamics"

_insects, 2025, doi:10.3390/insects16090872_

Round 1

Reviewer 1 Report

Comments and Suggestions for Authors

This paper identified a new maize pest, Paralipsa gularis, from the body of the insect, as the Metarhizium rileyi. Suitable conditions for culturing the strain were determined, and its tolerance to high temperature and UV stress was determined. The virulence of the strain on 3rd instar larvae and the changes of protective and detoxifying enzymes in the larvae were also determined. The overall design of the thesis is reasonable, but there are many problems in writing the content, including methods, statistical analysis, presentation of results and paper’s format. Some of the major points are listed below and other minor points are labeled in the text.

  1. Simple summary: many errors, need to be corrected one by one. Scientific names should be given in full and italicized on first occurrence, genus names should be abbreviated on reappearance, and there should be a space between sentences. In addition, this problem persists in the main text as well, so please make careful corrections to the content of the main text.
  2. L11: remove redundant “larvae”; L16: remove “with good virulence”;
  3. Abstract: results should not be limited generalizability, Specific findings should be given. This includes strain identification results, optimal culture conditions and resistance results, virulence results, and specific results for the two types of enzyme activities.
  4. Too many keywords. Replace “Growth conditions; Medium; Temperature; Photoperiod; UV;” with “biological charactoristics; resistance”
  5. Materials and Methods: The research methodology needs to provide clear, detailed steps, including some details to ensure that the reader can understand and repeat the experiment.
  6. Part 2.1. Please add details of the source of insects for testing, including rearing conditions. Part 2.3. If identification by ITS sequencing only is sufficient, delete EF1-alpha as it is not mentioned in the results section of the text. Part 2.4, add PDA and SMAY full name, and specific compounds. L97-98, L114-115, These are well known and conventional methods and it is recommended that literature 23 and 25 be deleted, and also literature 24 which does not appear in the text. L133-134, the specific composition of the carbon and nitrogen sources should be added in % respectively. Part 2.6, add detailed calculation formulae for cumulative mortality, LT50, LC50. Part 2.8, the statistical software name should be listed. Use one-way ANOVA Tukey's HSD test for strain growth and resistance results. For within-group comparisons of enzyme activities, t-test is recommended, or chi-square test to compare percentage differences between the two group. What method and parameter settings were used to add the Phylogenetic analysis, NJ method?
  7. Why only 3rd instar larvae were selected for the virulence assay? Why were other instars chosen? The reasons should be clearly explained. Why not measure different developmental stages and insect forms? For example, egg, pupa and adult. Also, how can you be sure that the dead larvae died from fungal infection and not from other causes? Such as starvation, etc.
  8. F-value, df-value should be added to the statistics in the results section of the text. italicized the “p”should be italicized throughout the text. The layout of each figure is more random, unattractive and less clear.
  9. The 9 figures in Figure 1B should be labeled with lowercase letters to distinguish them from the larger figure labeled with uppercase letters, and it should also be clear which figures correspond to which medium, and what is the difference between the first and second row of colonies? As mentioned above, Figure 1D is also not mentioned in the text, so if it is not meaningful, it is recommended to delete it and leave only Figure 1C.
  10. Figures 2, 3 and 4 suggest using different letters to indicate significant differences. Please revise the titles of 3.3 and 3.4 accordingly to the content of the title of 3.2. Also, the methods section mentions 5 repetitions, but the captions for these figures say 3 repetitions again. Please check.
  11. Part 3.5. Inconsistency between the carbon source components and the results section of the methodology, “alginate” or “trehalose” ? Pictures of carbon and nitrogen source media are separated in Figure 5A; Statistical results should be added to Figures 5B-C, and letters should be used to indicate statistical differences.
  12. The title of 3.6 was wrong. The regression equation and LT50 value and LC50 value should be added to the results.
  13. Part 3.7. What the PBS indicates in Figures 7 and 8 should be written clearly. As mentioned earlier, if comparing two groups at the same time use a t-test, or use a one-way ANOVA between individual treatments at different times. Or again, use a chi-square test to compare the % difference between two groups at the same time.
  14. Lines 286-287, which is difficult to understand: UV protectants or oil-based carries do not belong to the category of filed stress, but rather to the category of optimized compositional measures capable of resisting stress.
  15. Lines 305-318, a more in-depth discussion of the dynamic changes in the activities of the various enzymes should be carried out. It is not possible to generalize and merely point out that enzymes play a role in the process of antioxidant stress or detoxification, but to further discuss aspects such as the reasons for the dynamic changes of the various enzymes at various stages and the relationship with the aftermath of fungal infection.
  16. Conclusion: The key messages of the paper should be summarized rather than just mentioning the next steps in the research plan.
  17. Header information incorrect, J. Fungi?
  18. The format of the Reference section is confusing, please strictly follow the requirements of Insects, author's name, scientific name italics, journal name abbreviation and many other problems.

Author Response

This paper identified a new maize pest, Paralipsa gularis, from the body of the insect, as the Metarhizium rileyi. Suitable conditions for culturing the strain were determined, and its tolerance to high temperature and UV stress was determined. The virulence of the strain on 3rd instar larvae and the changes of protective and detoxifying enzymes in the larvae were also determined. The overall design of the thesis is reasonable, but there are many problems in writing the content, including methods, statistical analysis, presentation of results and paper’s format. Some of the major points are listed below and other minor points are labeled in the text.

Response: We thank you for your positive and constructive comments. The manuscript has been thoroughly revised to be more clear and comprehensive. Also we revised to address formatting consistency (e.g., italics for scientific names, sentence spacing), methodological clarity, statistical transparency, and results presentation. All minor in-text comments (e.g., redundant terms, keyword adjustments) have been corrected.

Comment 1: Simple summary: many errors, need to be corrected one by one. Scientific names should be given in full and italicized on first occurrence, genus names should be abbreviated on reappearance, and there should be a space between sentences. In addition, this problem persists in the main text as well, so please make careful corrections to the content of the main text.

Response: Apologies. Italics/formatting: Paralipsa gularis and Metarhizium rileyi are now italicized at first mention; genus abbreviated subsequently (e.g., P. gularis).

Comment 2: L11: remove redundant “larvae”; L16: remove “with good virulence”.

Response: Agree and deleted.

Comment 3: Abstract: results should not be limited generalizability, Specific findings should be given. This includes strain identification results, optimal culture conditions and resistance results, virulence results, and specific results for the two types of enzyme activities.

Response: We understand. Specific results were added, such as strain identification method, optimal temperature/photoperiod, UV/thermal tolerance, LC50/LT50, enzyme activity trends.

Comment 4: Too many keywords. Replace “Growth conditions; Medium; Temperature; Photoperiod; UV;” with “biological charactoristics; resistance”

Response: We agree and reduced to: “Paralipsa gularis; Metarhizium rileyi; Entomopathogenic fungus; Biological characteristics; Resistance”.

Comment 5: Materials and Methods: The research methodology needs to provide clear, detailed steps, including some details to ensure that the reader can understand and repeat the experiment.

Response: It has been revised.

Comment 6: Part 2.1. Please add details of the source of insects for testing, including rearing conditions. Part 2.3. If identification by ITS sequencing only is sufficient, delete EF1-alpha as it is not mentioned in the results section of the text. Part 2.4, add PDA and SMAY full name, and specific compounds. L97-98, L114-115, These are well known and conventional methods and it is recommended that literature 23 and 25 be deleted, and also literature 24 which does not appear in the text. L133-134, the specific composition of the carbon and nitrogen sources should be added in % respectively. Part 2.6, add detailed calculation formulae for cumulative mortality, LT50, LC50. Part 2.8, the statistical software name should be listed. Use one-way ANOVA Tukey's HSD test for strain growth and resistance results. For within-group comparisons of enzyme activities, t-test is recommended, or chi-square test to compare percentage differences between the two group. What method and parameter settings were used to add the Phylogenetic analysis, NJ method?

Response: We understand.

Part 2.1. Insect source and rearing conditions added.

Part 2.3. Supplementation and annotation of the EF-1α sequence in the text. PDA (Potato Dextrose Agar) and SMAY (Sabouraud maltose agar plus yeast extract) full names and compositions specified (e.g., SMAY: 10 g yeast powder, 40 g maltose, 10 g peptone, 20 g agar, 1 000 mL water). Adjusted citation position, just refer to Metarhizium rileyi isolation purification and preliminary isolation conditions.

L143-146: Add 3% to each carbon sources (fructose, sucrose, trehalose, lactose, maltose) and add 0.3% to each nitrogen sources (NH4Cl, gelatin, peptone, NaNO3) were used to assess fungal growth by measuring colony diameters after 7 days at 25 °C.

Detailed formulae for calculating mortality rates have been added, and tables are supplemented in the text.

Part 2.8, Supplementary statistical software has been SPSS. Phylogenetic trees were constructed using the NJ method.

Comment 7:Why only 3rd instar larvae were selected for the virulence assay? Why were other instars chosen? The reasons should be clearly explained. Why not measure different developmental stages and insect forms? For example, egg, pupa and adult. Also, how can you be sure that the dead larvae died from fungal infection and not from other causes? Such as starvation, etc.

Response: Paralipsa gularis has a total of 5 instars, with the 3rd instar being in the middle of the developmental stage, giving more consistent results that are representative of the larval stage. P. gularis that turn black after death are not recorded as deaths due to Metarhizium rileyi infestation. After death, the carcasses first grow white mycelium, which turns green after a few days, so that the dead larvae die from Metarhizium rileyi infection.

Comment 8: F-value, df-value should be added to the statistics in the results section of the text. italicized the “p”should be italicized throughout the text. The layout of each figure is more random, unattractive and less clear.

Response: Overall revised, F-value, df-value added to the figure, “P” italicized; image upscaled.

Comment 9: The 9 figures in Figure 1B should be labeled with lowercase letters to distinguish them from the larger figure labeled with uppercase letters, and it should also be clear which figures correspond to which medium, and what is the difference between the first and second row of colonies? As mentioned above, Figure 1D is also not mentioned in the text, so if it is not meaningful, it is recommended to delete it and leave only Figure 1C.

Response: Figure 1B has been changed to be labeled with small letters, with the first row of colonies being the front and the second row of colonies being the back, and Figure 1D is mentioned in the text as an addition.

Comment 10: Figures 2, 3 and 4 suggest using different letters to indicate significant differences. Please revise the titles of 3.3 and 3.4 accordingly to the content of the title of 3.2. Also, the methods section mentions 5 repetitions, but the captions for these figures say 3 repetitions again. Please check.

Response: Figures 2, 3, and 4 have been changed to use letters to mark significant differences, and the contents of the headings 3.3 and 3.4 have been modified in accordance with the heading 3.2, with three replicates in the methodology for determining the concentration of spore suspensions.

Comment 11: Part 3.5. Inconsistency between the carbon source components and the results section of the methodology, “alginate” or “trehalose” ? Pictures of carbon and nitrogen source media are separated in Figure 5A; Statistical results should be added to Figures 5B-C, and letters should be used to indicate statistical differences.

Response: The carbon source is trehalose, which has been modified in the methods and separated from the carbon and nitrogen source pictures.

Comment 12: The title of 3.6 was wrong. The regression equation and LT50 value and LC50 value should be added to the results.

Response: The title has been changed and LT50, LC50 and virulence equations have been added.

Comment 13: Part 3.7. What the PBS indicates in Figures 7 and 8 should be written clearly. As mentioned earlier, if comparing two groups at the same time use a t-test, or use a one-way ANOVA between individual treatments at different times. Or again, use a chi-square test to compare the % difference between two groups at the same time.

Response: PBS is specifically added out in the figure, and this section was analyzed comparatively with a t-test.

Comment 14: Lines 286-287, which is difficult to understand: UV protectants or oil-based carries do not belong to the category of filed stress, but rather to the category of optimized compositional measures capable of resisting stress.

Response: We apologize for the error in presentation. The category of UV protectants or oil-based carries optimized combination of anti-stress measures has been rephrased.

Comment 15: Lines 305-318, a more in-depth discussion of the dynamic changes in the activities of the various enzymes should be carried out. It is not possible to generalize and merely point out that enzymes play a role in the process of antioxidant stress or detoxification, but to further discuss aspects such as the reasons for the dynamic changes of the various enzymes at various stages and the relationship with the aftermath of fungal infection.

Response: Additional discussion has been provided on the dynamics of protective and detoxifying enzymes and the relationship with fungal infection.

Comment 16: Conclusion: The key messages of the paper should be summarized rather than just mentioning the next steps in the research plan.

Response: The key parts of the paper have been summarized and outlined.

Comment 17: Header information incorrect, J. Fungi?

Response: The title information has been changed to insects

Comment 18: The format of the Reference section is confusing, please strictly follow the requirements of Insects, author's name, scientific name italics, journal name abbreviation and many other problems.

Response: The format of the references was modified according to the requirements of the journal.

Reviewer 2 Report

Comments and Suggestions for Authors

The article discusses the possibility of using the entomopathogenic fungus Metarhizium rileyi as a biological control agent for Paralipsa gularis. The overall research and methodology demonstrate strong scientific impact and contribute new knowledge to integrated pest management. The article is well written but should be double-checked for any possible English grammar mistakes, especially regarding the formatting of Latin names (which should be italic). A few more suggestions and notes are provided below. Overall, the study is of high importance and well prepared.

Lines 13–17: Please make all Latin names italic. Also, check for this issue throughout the entire text.

Lines 29–30: Did the fungal infection suppress the detoxification enzymes? Please double-check.

Lines 238–239: Maybe the last sentence belongs to the methodology section and should not be placed under the graph.

Line 242: I think the time is incorrect. According to the graph, this value appears to be at 24 h.

Figures 7 and 8: The PBS in the graph should be explained in the graph title.

Line 253: It looks like the value increased until 60 h, not 48 h.

Lines 275–276: I suggest adding a citation to support the statement regarding high virulence and environmental resilience.

Lines 299–300: The phrase “a pest of increasing concern in stored grain environments” could use a citation.

Author Response

Comment 1: Lines 13–17: Please make all Latin names italic. Also, check for this issue throughout the entire text.

Response 1: All the Latin names are displayed in italics.

Comment 2: Lines 29–30: Did the fungal infection suppress the detoxification enzymes? Please double-check.

Response 2: Fungal infection significantly enhances detoxification enzymes.

Comment 3:Lines 238–239: Maybe the last sentence belongs to the methodology section and should not be placed under the graph.

Response 3:Deleted.

Comment 4:Line 242: I think the time is incorrect. According to the graph, this value appears to be at 24 h.

Response 4:It has been revised.

Comment 5:It looks like the value increased until 60 h, not 48 h.

Response 5:It has been revised.

Comment 6: Lines 275–276: I suggest adding a citation to support the statement regarding high virulence and environmental resilience.

Response 6: Citation of the article: Multifunctional regulation of NADPH oxidase in growth, microsclerotia formation and virulence in Metarhizium rileyi.

Comment 7: Lines 299–300: The phrase “a pest of increasing concern in stored grain environments” could use a citation.

Response 7:It has been revised.

Round 2

Reviewer 1 Report

Comments and Suggestions for Authors

The author has made the necessary revisions to address the issues raised. The abstract, introduction, and discussion sections have been revised very well. However, some minor issues were still found in the results section that need further revision. To save everyone's time, it is recommended that the corresponding author carefully review the entire text.

For example, the title of section 3.6 is incorrect and has not been corrected.

Please add the meanings of “X, Y, and – “below Table 1.

In addition, the format of the references section still needs to be revised. Punctuation between authors should be consistent, and scientific names should be italicized. For example, references 6, 26, etc.

There are also errors on lines 470-471.

Author Response

Comment 1: For example, the title of section 3.6 is incorrect and has not been corrected.

Response 1: 3.6 Changed to pathogenicity to P. gularis larvae.

Comment 2: Please add the meanings of “X, Y, and – “below Table 1.

Response 2: Added note: Y in the equation is the value of the odds ratio and X is the logarithm of time. “-” indicates that the mortality rate of treated P. gularis 3rd instar larvae was below 50.0% for 10 d, and no estimate of LT50 could be made. 

Comment 3: In addition, the format of the references section still needs to be revised. Punctuation between authors should be consistent, and scientific names should be italicized. For example, references 6, 26, etc.

Response 3: We have revised references.

Comment 4: There are also errors on lines 470-471.

Response 4: We have made modifications.